# Sensors Network as an Added Value for the Characterization of Spatial and Temporal Air Quality Patterns at the Urban Scale

**DOI:** 10.3390/s23041859

**Published:** 2023-02-07

**Authors:** Daniel Graça, Johnny Reis, Carla Gama, Alexandra Monteiro, Vera Rodrigues, Micael Rebelo, Carlos Borrego, Myriam Lopes, Ana Isabel Miranda

**Affiliations:** Centre for Environmental and Marine Studies (CESAM), Department of Environment and Planning, University of Aveiro, 3810-193 Aveiro, Portugal

**Keywords:** sensors, air quality, urban, sensors network, temporal patterns, sensors for smart cities

## Abstract

Within the scope of the Aveiro STEAM City project, an air quality monitoring network was installed in the city of Aveiro (Portugal), to evaluate the potential of sensors to characterize spatial and temporal patterns of air quality in the city. The network consists of nine sensors stations with air quality sensors (PM10, PM2.5, NO_2_, O_3_ and CO) and two meteorological stations, distributed within selected locations in the city of Aveiro. The analysis of the data was done for a one-year measurement period, from June 2020 to May 2021, using temporal profiles, statistical comparisons with reference stations and Air Quality Indexes (AQI). The analysis of sensors data indicated that air quality variability exists for all pollutants and stations. The majority of the study area is characterized by good air quality, but specific areas—associated with hotspot traffic zones—exhibit medium, poor and bad air quality more frequently. The daily patterns registered are significantly different between the affected and non-affected road traffic sites, mainly for PM and NO_2_ pollutants. The weekly profile, significative deltas are found between week and weekend: NO_2_ is reduced on the weekends at traffic sites, but PM10 is higher in specific areas during winter weekends, which is explained by residential combustion sources.

## 1. Introduction

Air pollution is recognized as the greatest environmental threat to human health [1], and risks such as an unhealthy diet and smoking have comparable impacts in relation to exposure to air pollutants [2,3,4]. This is particularly critical in urban areas, due to increased urbanization and multiple anthropogenic emission sources such as road traffic, residential combustion for heating, and industrial emissions [3,5]. To mitigate air pollution, it is necessary to define control strategies, which may include plans to reduce emissions and monitor their effects on air quality. Monitoring is generally carried out with reference monitoring stations used for regulatory purposes. However, due to the high cost of those systems, the number of reference monitoring stations is limited, preventing a detailed mapping and analysis of air quality in urban areas [3,6,7].

Low-cost sensors (LCS) are an emerging and quickly evolving technology area, commercially available in a wide variety of designs and capabilities [8]. Air quality sensors can be divided into six groups. For gases: electrochemical (EC), semiconductor/metal-oxide (MOS), photoionization sensors and non-dispersive sensors by infrared absorption (NDIR), and for particles: size-classifier sensors and light-scattering sensors [9,10]. The development of these sensors has revolutionized Air Quality Monitoring (AQM) since they have lower energy requirements and are less expensive compared to reference equipment while revealing an increasing quality. These sensors can then be installed in multiple locations where reference monitoring is not possible, providing a wider spatial coverage—one of the greatest advantages of LCS [11,12]. Therefore, these devices allow monitoring of larger areas and provide support to policymakers to implement strategies for improving air quality [3,6].

There are still some challenges associated with these technological devices, such as the lack of data to compare sensors and sensor systems with each other and with reference equipment; thus, uncertainties are still poorly characterized [13]. Sensors are sensitive to developing zero drift and ageing effects, which affect the calibration and can lead to systematic errors [9]. Moreover, difficulties are also linked to the high variability in the performance of similar sensors and the variations associated with different meteorological conditions and emission environments. Nevertheless, recent progress highlights the potential of this technology. The European Commission (EC) submitted a proposal to revise the Ambient Air Quality Directive as part of a ‘zero pollution’ package. The proposal includes new methods to measure concentrations or deposition levels of pollutants, such as in-situ sensors [14]. The Forum for Air Quality Modelling (FAIRMODE), coordinated by the European Commission Joint Research Centre (JRC), has created a specific working group on low-cost sensors which is developing new methodologies to combine sensors networks with modelled data and official measurements [15].

Several projects and studies have been carried out with networks of monitoring stations with sensors with different objectives, such as: comparing with reference methods, analyzing air quality in urban areas and possible air pollution hotspots, analyzing the dispersion of pollutants, validating atmospheric models, analyzing human exposure, and informing and engaging communities [6]. Additionally, studies (e.g., Borrego et al., 2018 [2] and Castell et al., 2016 [11]) have compared the performance of air quality sensors with reference measurements, exhibiting the sensors’ ability to perform measurements within the range of errors of indicative measures. Several communities and web platforms are currently available, aiming to bring together communities in a specific neighbourhood, city, region or country [16], such as the Aveiro STEAM City project (https://www.uia-initiative.eu/en/uia-cities/aveiro, accessed on 23 October 2022) [17]. This project aims at creating a smart city to implement and enhance policies focused on urban mobility, energy and the quality of the environment. For that, a network of monitoring stations, both air quality and meteorological, was installed in the city of Aveiro, Portugal. This network provides crucial information to communicate and raise citizens’ and policymakers’ awareness of the environmental challenges of the city. Besides, the city has only a reference air quality monitoring station mainly influenced by traffic, which has a restricted spatial representativeness of a few hundred meters [18]. Thus, the installed network will contribute to overcoming this lack of spatial representativeness.

The main objective of this paper is to present a comprehensive air quality assessment, based on a low-cost sensors network. The main findings of this paper will contribute to identifying the air quality temporal and spatial patterns of the city and to assess the performance of the installed network. The innovative nature of this paper relies on its case study demonstration in the city of Aveiro. The environmental observations discussed in this paper, acquired from a low-cost sensors network, which is embedded in the STEAM-City project, highlight a unique process of citizens’ engagement in multiple areas of society (e.g., environmental challenges, jobs creation, technological challenges) based on a STEAM approach. The outcomes of this study can assist citizens, as a learning tool, helping them to know the areas and periods with the highest levels of air pollution. Citizens, on the other hand, will be more aware of the data available thanks to the combination of both this unique process of citizen engagement and the data analysis provided in this paper. To achieve the main goal of this study, the manuscript was structured into four main steps with specific objectives: (1) Evaluate the performance of the sensors monitoring stations; (2) Characterize the air quality in the city of Aveiro; (3) Analyse the temporal and spatial patterns of the sensors data for all pollutants; (4) Evaluate the representativeness of the sensors network through clustering analysis.

## 2. Materials and Methods

### 2.1. Study Site

The city of Aveiro is the headquarters of the municipality of Aveiro with 80,880 inhabitants (2021 Census) [19]. It is a coastal city in the central region of Portugal and it is located on the outskirts of the Ria de Aveiro (Figure 1), surrounded by marshy areas and rural areas [20]. The city is at a low-lying altitude and is mainly flat [21,22].

Aveiro has a Mediterranean climate and, according to the Köppen classification, temperate wet, with a dry and mild summer (Cbs), with a strong maritime influence [3,23,24]. The average annual temperature is about 15 °C and the highest average monthly temperature values occur in the summer months (July, August and September) [25].

In the scope of the STEAM City project, a sensors network with nine sampling sites was installed, with air quality monitoring sensors, and two of the sites had a meteorological station. Figure 1 shows the location of the monitoring stations with the two locations measuring meteorological data highlighted in blue and the reference air quality monitoring station of Aveiro, Ílhavo and Estarreja marked in red.

The localization of the sensors stations was subject to specific conditions to ensure their proper functioning and monitorization of different parts of the city. These conditions include:Electricity and communication access;Adequate security measures;Least possible exposure to meteorological elements;Avoid proximity to sources of air pollution, such as chimneys and exhaust vents, to avoid sensor saturation and inaccurate measurements of ambient air;Installation preferably in buildings owned or managed by the municipality;Proximity to main avenues, highways, parking lots and tourist areas.

Due to these constraints, the sensors stations were installed on the top or on the side of buildings at heights ranging from 2 to 15 m.

The air quality and meteorology monitoring stations were installed between April and June 2020, except for the station at the train station, which was installed in November 2020. Considering the installation dates of the sensors stations, in this study the period analyzed was from June 2020 to May 2021.

Regarding the conventional air quality monitoring, the region of Aveiro has three stations (Figure 1 in red) from the Portuguese reference/official network: Aveiro (urban environment and traffic influence), Ílhavo (suburban environment and background influence) and Estarreja (suburban environment and background influence). The three monitoring stations are automatic and use the beta-attenuation method to measure PM10/PM2.5 concentrations, chemiluminescence to measure nitrogen dioxides, and ultraviolet photometry for ozone. However, only the Aveiro air quality reference station exists in this study area. This station, located in a school, near a main avenue, has been in operation since 2003, measuring concentrations of PM10, NO_2_, NO, NOx and CO, with data available on the Portuguese Environmental Agency (APA) website [26], and will be used for comparisons with the sensors data.

In 2020, the area combining Aveiro and Ílhavo had, on most days, good air quality (198 days) and very good (88 days) [27]. However, the impact of the lockdown measures imposed by the Portuguese government in 2020 due to the COVID-19 pandemic led to a consequent generalized improvement in air quality, resulting in an atypical year [27,28]. During 2021 the pandemic also had an impact on air quality although less expressive compared to 2020 [27].

### 2.2. Sensors Stations Monitoring Equipment

The air quality stations include sensors to measure PM10, PM2.5, O_3_, NO_2_ and CO, and several other components to guarantee a stable and acclimatized air inflow at a constant temperature, namely, a flow meter, internal temperature sensors and other auxiliary electronic components to process and transfer data to the servers. The components are encased in a hermetic box and have one inlet of ambient air and one outlet. Inside the station, there are two independent circuits of air, one for particles and one for gases.

The characteristics of the sensors used are shown in Table 1.

The calibration of sensors refers to two processes. First, to ensure the quality of the gas sensors measurements, they were calibrated with reference gases before being installed in monitoring locations, so establishing a relationship between indicative measurements and standard (reference) measurements, estimating the parameters of the calibration function [29]. Second, all the sensors that were assembled on the sensors stations are subjected to outdoor ambient air measurement conditions (after being installed in the stations) and compared with measurements from reference instruments in co-location, to validate their performance when exposed to outdoor meteorological conditions [30]. This calibration was made by the supplier company, which has an ISO 17025 accredited laboratory for air quality testing and has a mobile air quality station. The sensors were calibrated against reference monitoring methods, as defined in Directive 2008/50/EC (UE 2008) [18] under factory conditions [31].

Before the equipment was delivered, a calibration was performed with its mobile station to measure and adjust the sensors data. With this calibration, it was also possible to assess the ability of the sensors to respond to external stimuli and to verify the variation in exposure to meteorological conditions. During the study period, the company ensured equipment maintenance and data surveillance.

The meteorological stations were installed on a mast measuring about 1.5 to 2 m (Figure 2) on the buildings’ top at about 10 m. Therefore, the measurement height of the meteorological parameters is still located within the urban canopy layer, and the obstacles can induce perturbations on the flow structure and thermodynamic properties (e.g., advective effects, leeward effects). When the measurements showed abnormal changes in their magnitude there were adjustments in the sensor’s sensitivity via wireless communication.

For communication of the collected data, the installed monitoring stations use Ethernet and LoRa technology in a modular way and support other possible communication modules, WiFi, NB-IoT and 3G/4G technologies.

### 2.3. Data Collection and Validation Procedure

The data quality was assessed from the beginning of the installation of the stations and involved a pre-processing (see Figure 3) with careful sorting of the data, through the analysis of the time series, removing outliers that coincided with periods of calibration and/or maintenance of the equipment and with technical problems in the monitoring stations. This analysis and data removal considered the data provided by the field surveys, which made it possible to understand how the particularities of each station justify certain different patterns or behaviour in data. In addition, the meteorological parameters and atmospheric pollution events that occurred were considered in this analysis and data removal decisions. To facilitate the analysis and comparison of the pollutant concentrations with the reference stations, daily averages were calculated for every pollutant and station only for the hours with a minimum of 75% of data on each hour.

### 2.4. Statistic Evaluation Indexes

The evaluation of the sensor’s performance was done through the comparison of the measurements of the sensors stations (S) against the air quality reference station of Aveiro (R) at a time (t). Three different metrics were used, where n represents the total number of observations and σ represents the standard deviations:

Root Mean Square Error (RMSE)
(1)$$\mathrm{RMSE}=\sqrt{\frac{\sum_{t=1}^{n}{\left(S_{t}-R_{t}\right)}^{2}}{n}}$$

Normalized Mean Bias (NMB)
(2)$$\mathrm{NMB}=\frac{\sum_{t=1}^{n}\left(S_{t}-R_{t}\right)}{\sum_{t=1}^{n}R_{t}}$$

Pearson correlation coefficient
(3)$$r=\frac{1}{n-1}\sum_{t=1}^{n}\left(\frac{S_{t}-\bar{S}}{\sigma_{S}}\right)\left(\frac{R_{t}-\bar{R}}{\sigma_{R}}\right)$$

The evaluation of the sensors stations performance was based on statistical parameters that are widely used and accepted for data comparisons [2,3,6]. The RMSE indicates the magnitude of the error and the NMB provides the magnitude of differences between sensors and reference values averaged over the whole sampling period. The r (Pearson) measures the strength and direction of a linear relationship between two variables [3].

### 2.5. Graphical Visualization

To easily visualize the state of air quality, AQI were produced, using Python with the libraries pandas, datetime, NumPy and matplotlib [32], from the data compiled and validated for each sensors station (Figure 3). Graphs were prepared with the AQI, presented in Section 3.3, for each monitoring site that presents the frequency of air quality indexes, according to the APA (Portuguese Environmental Agency) standards, for the entire data period considered in the analysis.

The calculation is based on the daily arithmetic averages for PM10 and PM2.5 and maximum values for NO_2_ and O_3_, only for days with a minimum of 75% of daily data. The AQI value is then translated into a colour scale divided into five classes, from “Very Good” to “Bad” and also “No data”, which is represented by the pollutant that obtained the worst classification. These classes are based on the knowledge of the effects of these pollutants on human health and the values recommended by the World Health Organization (WHO) [33].

To compare the seasonal daily, weekly and monthly profiles between sensors stations, to evaluate sources based on wind speed and direction (PolarPlot and PollutionRose) and to statistically compare the sensors stations with the reference station, graphs and tables were made using R programming and the OpenAir library [34,35]. With the same library, dendrograms were plotted to make a cluster analysis of the sensors and reference stations using the complete-linkage method, which is a hierarchical clustering technique that uses the farthest distance between data pairs to determine inter-cluster distances [36].

## 3. Results and Discussion

### 3.1. Validation Procedure

Despite regular maintenance and calibration, during the measurement period, several problems affected the data collection efficiency of the monitoring stations, such as:Problems in the proper functioning of the sensors due to interference from temperature and humidity extremes (condensation inside the station and/or in the sampling sockets);Deviations in concentrations measured by sensors and need for adjustments of the baseline;Malfunctions of internal components in the monitoring stations (filling of filters, rupture of filter door and problems in the thermal regulation of the station during meteorological extremes);Communication failures with data storage systems;Power failures resulting from problems in the electricity grid and lightning electrical discharges.

Table 2 shows the overall efficiencies for each air quality sensor for the entire study period, after data processing and validation, considering the identified problems affecting the monitoring stations and the requirements of the Framework Directive 2008/50/CE.

To ease the analysis, colour gradients were applied to the results of Table 2 on a scale ranging from green to red, with yellow or brown in between according to the statistical variable.

Among the installed stations, almost all of them had an efficiency above 75% for each pollutant. The exception is the Chapel, whose efficiency for NO_2_ and O_3_ was only 55% and 53%, respectively, due to continuous technical problems affecting this station.

It is possible to identify and explain some causes for periods in which failures and common or particular pollution events occurred at each station. In Appendix A, it is possible to verify that the periods with the greatest data failures are concentrated in the autumn and winter periods and mainly in the electrochemical sensors of the pollutants O_3_, NO_2_ and CO. The main reasons for this situation are related to the lower or higher temperatures, greater amounts of precipitation and relative humidity at these times that interfere with these sensor’s measurements.

### 3.2. Comparison between Measurements from the Sensors Stations and the Reference Station

Table 3 shows the results of the comparison between the sensors stations measurements with the data from the reference station of Aveiro for the pollutants measured at this station (NO_2_, PM10 and CO), using the validation metrics previously identified. This analysis was not done for PM2.5 and O_3_ because the reference stations that monitor these pollutants are far from the influence of the city of Aveiro (>5 km).

To facilitate the analysis, colour gradients were applied to the results of Table 3 on a scale ranging from green (fewer errors/high correlation) to red (more errors/less correlation), with yellow/brown in between according to each statistical metrics interval.

This comparison is being made with an urban traffic station, representative of a very small area, and as such large deviations are expected as locals move away from this station. However, results show that even some stations near the reference station can have less correlation than faraway sensors stations in some pollutants (e.g., CETA compared to Morgados for NO_2_ and CO).

Results for NO_2_ show an RMSE in a range of 10.53–30.35 µg/m³ and positive NMB (0.151–1.336 µg/m³) in all the stations, indicating that the sensors had higher values than the reference station. The correlations with the reference station are in the range of 0.11–0.67. Most of the sensors stations have low correlations or show no correlation compared to the Aveiro reference station. The University sensors station had an extremely low correlation (0.11), probably because it is at a higher altitude in relation to the roads and may not receive such a direct and intense influence from road traffic peaks, typically visible in concentrations of NO_2_ [4]. The sensors station at CETA was the one that presented the highest correlation (0.67) for NO_2_ despite being at a great distance from the reference station, probably due to the high influence of the highway on the CETA station, which will likely have variations in road traffic closely related to traffic peaks close to the reference station. At the firefighter’s headquarters sensors station, the RMSE and NMB are positive and high, indicating that concentrations are consistently higher at this location.

For PM10, RMSE is in a range of 12.59–21.07 µg/m³ and NMB is generally negative (−0.36–0.05 µg/m³), except for the station in Museum (0.05 µg/m³), indicating that the sensors had generally lower values than the reference station. The correlations with the reference station are in the range of 0.56–0.73. Compared to NO_2_, PM10 has higher correlations (0.56–0.73) showing that throughout the city the variation in PM10 concentrations has a moderate correlation. Concerning CO, it was found that the correlations are high, except for the Firefighters Headquarters (0.27) and the Chapel (0.31). The RMSE of these two stations are also quite high; therefore, there are high fluctuations in the measured concentrations. In firefighters, as in NO_2_, this situation is due to the scheduled start of vehicles from the parking lot, discussed in Section 3.4.

For CO, RMSE is in a range of 100.44–397.16 µg/m³ and NMB in a range of −0.86–0.149 µg/m³. The correlations with the reference station are good in general, with seven sensors stations with correlations between 0.8 and 0.9, except for the Firefighters Headquarters and Chapel sensors stations (0.27 and 0.31, respectively), demonstrating that these two locations may have different emission sources or different temporal patterns of emission. At the Chapel sensors station, very high and regular concentration peaks were identified in the CO concentration of unknown origin (Appendix A) which caused an increase in the RMSE and low r values.

### 3.3. Air Pollution Extremes

During the measurement period, some extreme atmospheric pollution events from anthropogenic and natural sources negatively influenced the air quality in the city of Aveiro and affected the measurements of the sensors network.

The most critical atmospheric pollution event was the rural fire that broke out on the 8 September 2020 in the municipality of Albergaria-A-Velha, located about 15 km east/northeast of the city of Aveiro. Due to conditions of high stability of the atmosphere and east wind during the morning, it produced a well-defined column of smoke that affected the air quality of the city of Aveiro (Appendix A).

All monitoring stations reached, in a short period, high concentrations of PM10, PM2.5 and CO (Appendix A). In this event, the Aveiro air quality reference station reached a concentration of PM10 of 1000 µg/m³, the limit of detection, and two stations reached this concentration value (Morgados and Museum). The other sensors stations did not exceed concentrations of around 500 µg/m³, which corresponds to the maximum concentration in the measurement range of the particle sensors equipped in the stations (Table 1).

Another common air pollution episode occurs in winter due to residential combustion for heating. Much of domestic heating is still done using mainly wood combustion, which produces mainly PM10, PM2.5 and CO [37]. Similarly, Danek and Zareba, 2021 present a study assessing the air quality in the city of Krakow, Poland, using LCS data. The main findings of this study highlight that air pollution episodes during winter in Krakow are mainly caused by the use of solid fuel for residential heating in the neighbouring cities [24,38]. Usually, these episodes occur during cold nights, with high atmospheric stability and especially when thermal inversions occur in the lower atmosphere, the ideal conditions to reduce the dilution of pollutants in the atmosphere. This pollution episode occurred mainly in the first half of January, associated with a “situation of blockage of the west current and, consequently, meridional flow with the transport of a mass of polar air to the Iberian Peninsula” [39].

### 3.4. Temporal Analysis

In this subsection, the daily, weekly and seasonal profiles of the pollutant concentrations are presented and analysed to better characterize the different temporal patterns found in Aveiro city (Figure 4, Figure 5, Figure 6, Figure 7 and Figure 8).

Regarding PM10 and PM2.5 (Figure 4 and Figure 5), the daily, weekly and seasonal variations show similar patterns for all the sensors sites. Particle concentrations are significantly higher at night, especially in winter, but also in spring and autumn. In winter, this increase may be caused by emissions from residential combustion for domestic heating and boosted by the high stability of the atmosphere and lower temperatures at night, conditions that occurred more sharply in January [37]. In the autumn months and in March there may also be some contribution from this type of source due to the existence of relatively low temperatures during some nights. During the afternoon, a general reduction in concentrations is visible due to greater instability of the atmosphere in this period, induced by daytime heating and by the increase in the intensity of the sea breeze, causing a greater dispersion of particles in the atmosphere [40]. The peak registered in autumn is due to the episode of forest fires that impacted the air quality in Aveiro on 8 September 2020. In terms of monthly variation, there is a decrease from spring to summer (with April, May, June and August having the lowest concentrations).

The reference station’s monthly and weekly patterns are similar, but daily patterns in spring and summer have some differences. In spring the reference station has higher concentrations than the sensors stations during the day, and in summer it has higher concentrations at night. This indicates that the group of sensors installed in the city also exhibit different patterns at time scale and the reference station is not representative of the city area.

Figure 6 presents CO concentrations, which are higher mainly at night in the coldest months, potentially due to residential combustion. In autumn and spring, the CO values also reach higher values, mainly at the beginning of the night due to the greater stability of the atmosphere that reduces the dispersion of pollutants, similar to PM10 and PM2.5 [41].

There are also two specific peaks of CO in the morning between 8 am and 9 am and the late afternoon, between 7 pm and 8 pm. These peaks are most likely due to emissions from commuting road traffic between jobs and family homes that occur mainly at these times. These patterns are also significantly visible in NO_2_ concentrations (Figure 7), since both pollutants are traffic-related.

In the variations of CO concentrations throughout the week, concentrations are higher on weekends, especially in winter (except at Chapel station), contrary to what happens in autumn. This behaviour is also visible in the concentrations of PM10 and PM2.5 (Figure 4 and Figure 5) and may be explained by the fact that a greater number of inhabitants stay in their homes on weekends and need to heat the homes using combustion processes, for an extended period. The Chapel station is an exception, with higher concentrations on Sundays, in autumn and in winter. This behaviour is only evident in the CO and can be explained by the work of a nearby automotive workshop or other specific activities nearby.

The profiles are similar between the sensors stations and the reference station, but in June and July the concentrations of the reference station are much lower. In general, during summer, the concentrations of PM10 and CO are generally lower (in the absence of wildfires), increasing in winter, autumn and spring.

In Figure 4, Figure 5 and Figure 6, is possible to verify a different pattern in the daily summer profiles for the Museum station with two high concentrations peaks of PM10, PM2.5 and CO between 12 and 1 p.m. and 8 and 9 p.m., coinciding with the lunch and dinner periods.

We can attribute a source to these peaks by analyzing Figure A1, for CO, for example, where it is possible to see that in summer the high concentrations came mainly from the Northwest, where there is an area with restaurants that, in the summer, typically do grilled meals in charcoal.

Concerning NO_2_ (Figure 7), one of the main pollutants resulting from road traffic, it is possible to verify that the Firefighter’s station has a positive concentration deviation of about 20 µg/m³ concerning the other stations during the winter. Regarding the daily patterns, this sensors station has the highest daily peaks in spring and winter (in the early morning and late afternoon) which are mainly due to a daily routine that is related to turning on all parked vehicles as a way of verifying their operability that takes place every morning and can be proven by the PolarPlot of the Figure A2.

In a weekly analysis, it appears that, in general, NO_2_ concentrations are lower on weekends because there are lower road traffic levels. In summer, NO_2_ concentrations remain practically the same throughout the week. The Museum, Firefighters and CETA stations show the highest value, as they are under great influence of road traffic. The reference station has similar patterns and concentrations when compared with the sensors stations, except in the spring when the concentrations of the reference station are lower.

Finally, regarding O_3_ (Figure 8), daily typical daily patterns are registered in the majority of the sites with ozone consumption during the night and its formation during the day due to the role of solar radiation in the photochemical processes of ozone [40]. The patterns are practically the same in all season stations; only the concentration magnitudes vary. In general, sensors stations follow the patterns of the reference station of Estarreja but have higher concentrations, especially at night.

### 3.5. Spatial Analysis

To complement the previous analysis, which focused on time variation and temporal profiles, a spatial analysis was also performed to understand the variability existent over the city and to draw conclusions regarding the representativeness of the sensors network.

This analysis starts with Figure 9, showing the AQI for each monitoring site, which consists of the frequency of air quality indices (according to APA standards) for the entire data period considered in the analysis.

It should be noted that failures in measurement and/or communication of sensors stations can occur in peak concentration of air pollution extremes, meaning, sometimes, a lower frequency of worse AQIs (which happened in the Morgados sensors station during the wildfire of 8 September 2020).

Figure 9 shows that, besides some similarities found in the AQI frequency over the different monitoring sites, there is also a significant variability at the spatial level. This variability can be associated with the pollutants (responsible for the AQI colour) or even with the pollution level. To investigate it and better characterize this variability, dendrograms and cluster analysis were applied and presented in Figure 10.

This clustering analysis allows us to identify the similarities that exist among the sensors network for PM (PM10 and PM2.5), with two sub-groups of sensors that exhibit even higher similarity: (1) Train station, CETA, Chapel, Library and CC and (2) Morgados, Firefighters and UA. It also highlights that none of the reference stations are representative of this monitoring sites, which confirms the importance of this type of monitoring network around the city. The results for CO indicate that, except for the Chapel site outlier, there is a similarity among the different locations, with four main sub-groups identified.

Regarding NO_2_, the figure confirms the non-representativeness of the reference stations and the existence of three sub-groups of stations with similar behaviour.

For O_3_ the conclusions are different: the background reference station of Estarreja can characterize the levels of O_3_ measured in the majority of the sensors stations, with the exception of three locations.

## 4. Conclusions

Advances in science and technology make it possible to build sensors and components that are increasingly cheaper and of higher quality, making them a very important tool in the future of air quality monitoring.

In this context, the Aveiro STEAM City project created a network composed of nine monitoring stations with air quality sensors and two weather stations, distributed in selected locations within the city of Aveiro, Portugal. This network provides data to the population through online platforms and allows the detection and mapping of hotspots in terms of air pollution, also helping policymakers to take the necessary measures.

The air quality data recorded during a one-year period (from June 2020 to May 2021) shows that the air quality in Aveiro is, in general, good, despite having some periods of medium, poor and bad quality, and that the sensors responded adequately, providing measurements in a range consistent with the reference stations located within the city.

The temporal patterns indicate that there are strong influences from commuting road traffic, from residential combustion for heating purposes during winter, and more particularly in some places by point sources such as restaurants associated with tourism, like museums. Episodes of critical pollution episodes were also identified, with high concentrations of pollutants (mainly PM10 and PM2.5), exceeding the guidelines and limit values for the protection of human health, namely on 8 September 2020, when forest fire events surrounding the city impacted the urban area, and in January due to low temperatures and residential combustion.

In terms of spatial analysis, supported by the clustering analysis approach, it was found that three distinct areas/groups of sensors exist, with similarities in terms of time correlation and magnitude of pollutant concentration, which suggest that a network with distinct monitoring sites is required but the number of sensors could be reduced and optimized.

In summary, this study highlights that the reference (traffic) station located in Aveiro is not valid for characterizing the air quality in Aveiro city, in particular the distinct temporal patterns, air pollution hotspots, and for building the overall map of air quality in Aveiro city. This information is particularly valuable in supporting decision-making, related to urban planning and transport management in the city, to implement measures to improve air quality. In addition, it allows the population to be alerted to the risk of exposure in certain areas and time intervals.

It also appears that air quality monitoring networks with sensors are versatile and useful tools for monitoring air quality in urban areas. The reliability of the sensors is always different (lower) from that of the reference stations. However, as discussed in the introduction, the cost of these devices is much lower, allowing wider spatial coverage of monitoring networks. Nevertheless, air quality monitoring sensors should not be seen as a replacement for reference stations, but as a tool to complement those.

## Figures and Tables

**Figure 1 sensors-23-01859-f001:**
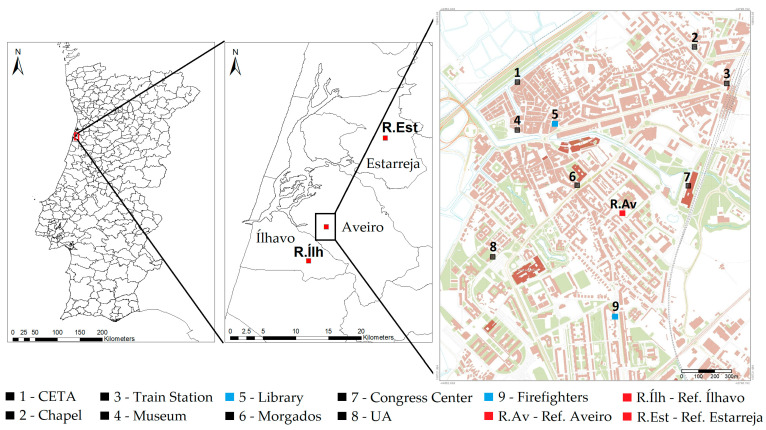
Location of the STEAM City air quality sensors stations (black and blue), meteorological stations (blue) and the reference air quality stations (red).

**Figure 2 sensors-23-01859-f002:**
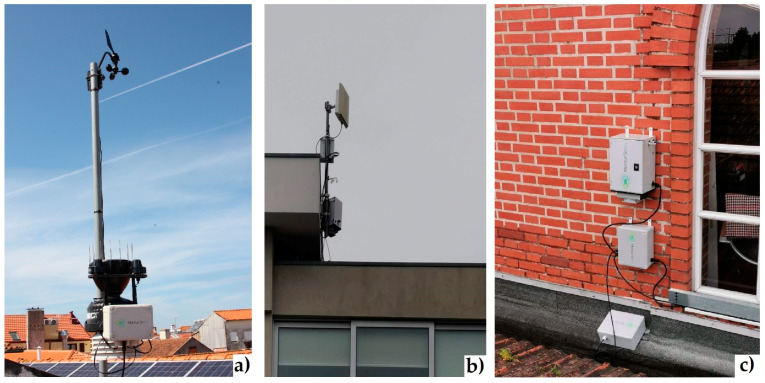
(**a**) Meteorological station at the Municipal Library (**b**), Air quality sensors stations in the University of Aveiro (**c**), and in the Congress Centre.

**Figure 3 sensors-23-01859-f003:**
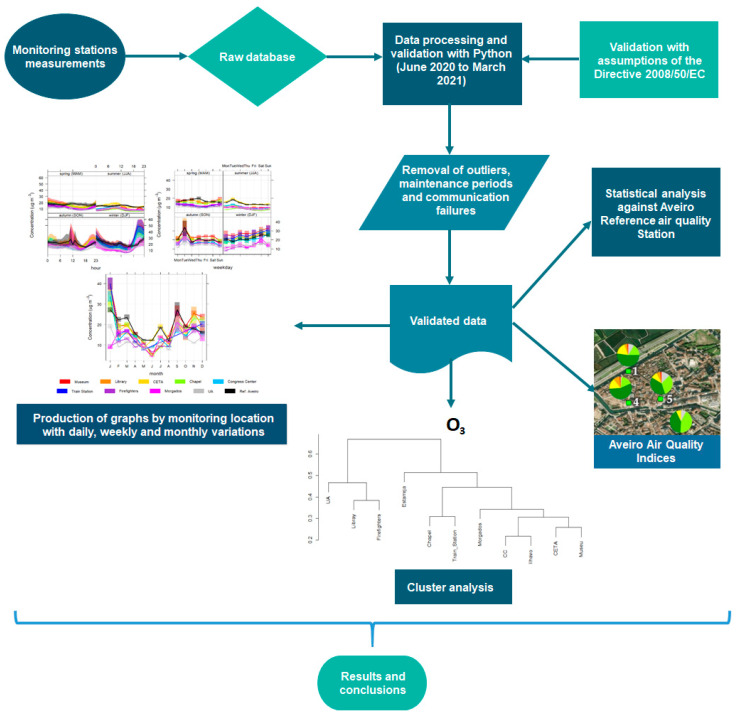
Scheme of the steps of the pre-processing and the outputs.

**Figure 4 sensors-23-01859-f004:**
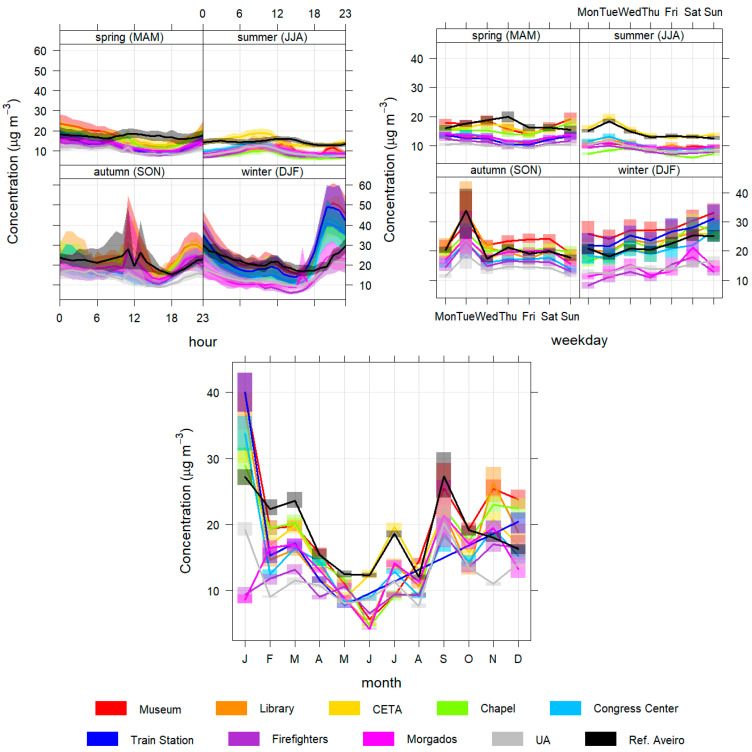
Daily (**top left**), weekly (**top right**) and monthly (**bottom**) profiles of PM10 for all monitoring stations.

**Figure 5 sensors-23-01859-f005:**
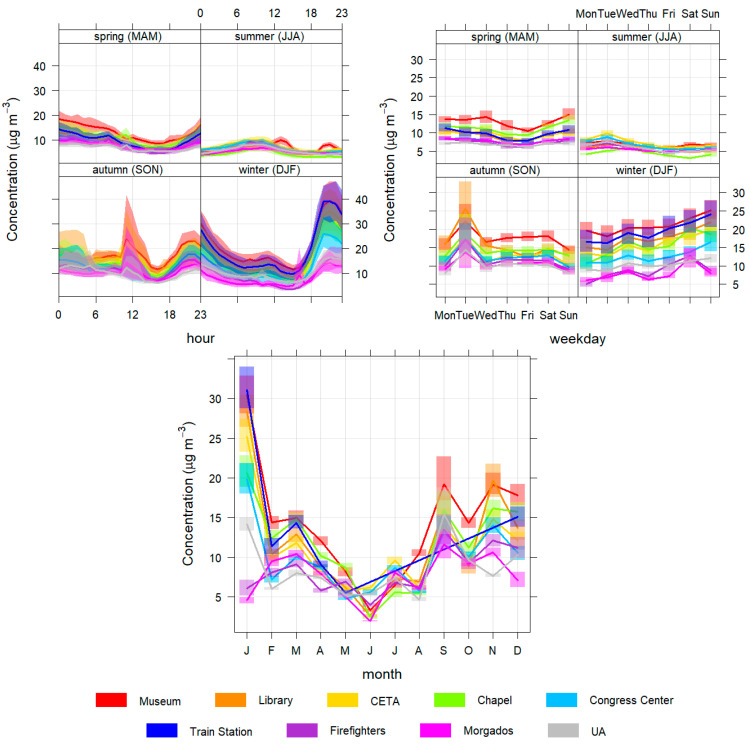
Daily (**top left**), weekly (**top right**) and monthly (**bottom**) variations of PM2.5 for all monitoring stations.

**Figure 6 sensors-23-01859-f006:**
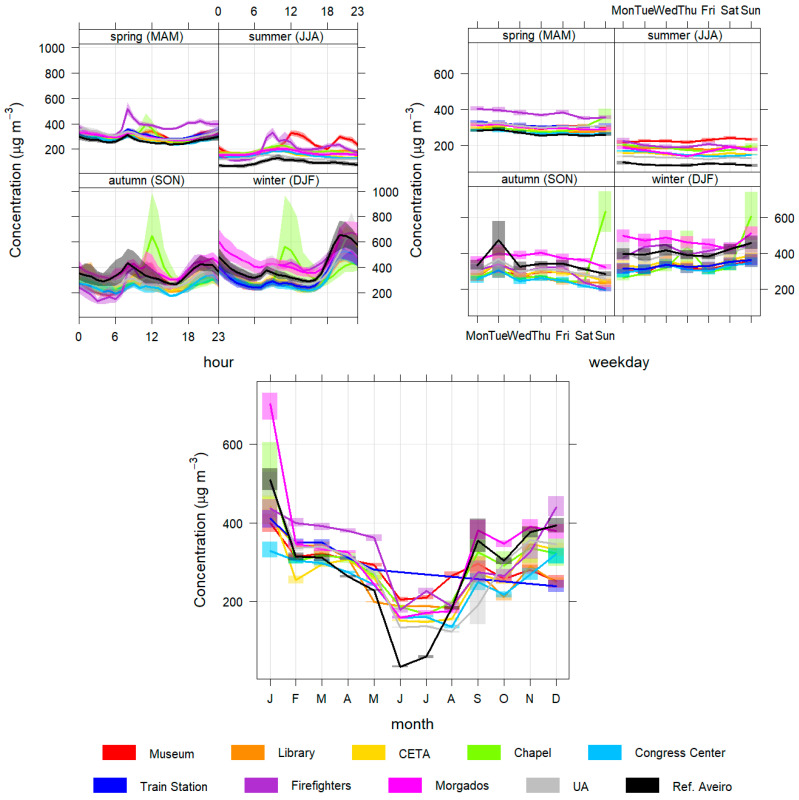
Daily (**top left**), weekly (**top right**) and monthly (**bottom**) CO variation for all monitoring stations.

**Figure 7 sensors-23-01859-f007:**
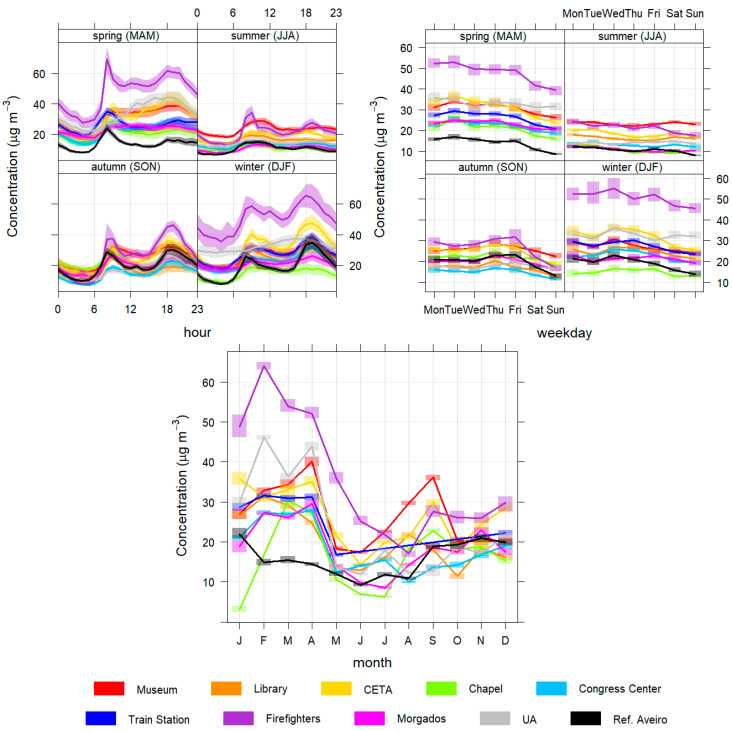
Daily (**top left**), weekly (**top right**) and monthly (**bottom**) variation of NO_2_ concentrations for all monitoring stations.

**Figure 8 sensors-23-01859-f008:**
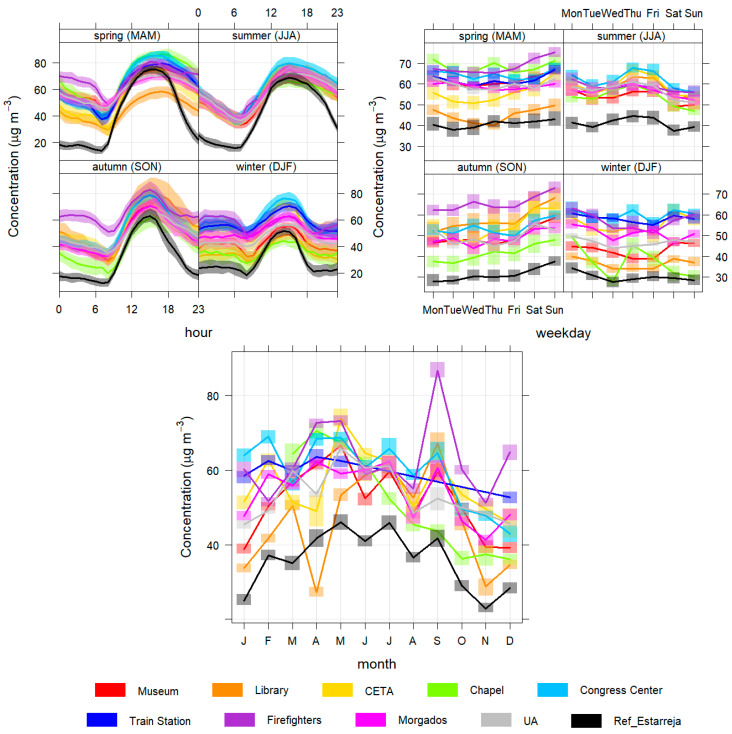
Daily (**top left**), weekly (**top right**) and monthly (**bottom**) variation of O_3_ concentrations for all monitoring stations.

**Figure 9 sensors-23-01859-f009:**
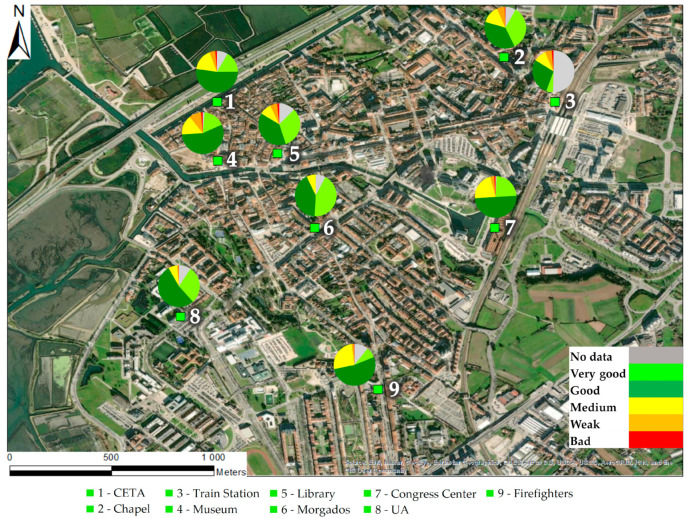
Distribution of the frequency of air quality indices in the city of Aveiro for the period between June 2020 and May 2021.

**Figure 10 sensors-23-01859-f010:**
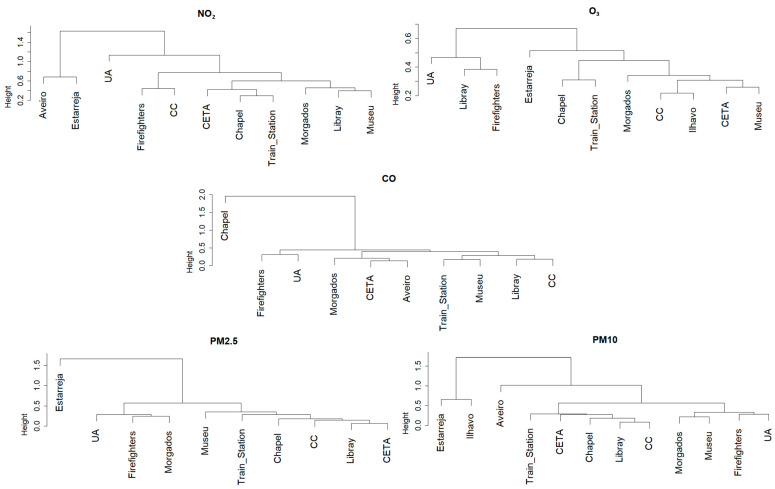
Cluster analysis for all pollutants and monitoring stations using the complete-linkage method.

**Table 1 sensors-23-01859-t001:** Main characteristics of sensors implemented in the network.

Pollutant	Technology	Model	Range (µg/m³)	Accuracy	Time Resolution (Minutes)
PM10	Light scattering	Gassensor	0–500	25% at 50 µg/m³	5
PM2.5	Light scattering	Gassensor	0–500	±25 µg/m³	5
O_3_	Electrochemical Sensor	Alphasense-AH	5–500	±25 µg/m³	15
NO_2_	Electrochemical Sensor	Alphasense-B42F	5–500	±15 µg/m³	15
CO	Electrochemical Sensor	Alphasense CO-AF	100–15,000	±55 µg/m³	15

**Table 2 sensors-23-01859-t002:** Data efficiency (%) related to each monitoring station by pollutant for the time interval considered in the analysis.

Stations	Environment	Influence	Efficiency (%)				
			PM10	PM2.5	NO_2_	O_3_	CO
1—CETA	-	-	92%	92%	93%	73%	95%
2—Chapel	-	-	92%	92%	55%	53%	91%
3—Train Station	-	-	99%	99%	99%	95%	99%
4—Museum	-	-	93%	93%	95%	94%	98%
5—Library	-	-	88%	88%	81%	78%	84%
6—Morgados	-	-	76%	76%	69%	76%	69%
7—Congress Centre	-	-	99%	99%	91%	93%	99%
8—University of Aveiro (UA)	-	-	86%	86%	67%	86%	84%
9—Firefighters Headquarters	-	-	76%	76%	76%	74%	81%
Ref. Aveiro	Urban	Traffic	97%		100%		100%
Ref. Ílhavo	Suburban	Background				50%	
Ref. Estarreja	Suburban	Background		98%		97%	

**Table 3 sensors-23-01859-t003:** Statistical analysis (RMSE, BIAS and r (Pearson)) for the STEAM City sensors stations compared to the Aveiro air quality reference station.

	NO_2_								
	Library	Firefighters	Chapel	Congress Centre	CETA	Train Station	Morgados	Museum	University
RMSE	13.51	30.35	10.53	11.79	16.06	15.62	12.96	17.89	21.37
NMB	0.33	1.336	0.197	0.151	0.644	0.625	0.157	0.676	0.54
r (Pearson)	0.43	0.4	0.56	0.5	0.67	0.57	0.42	0.41	0.11
	PM10								
	Library	Firefighters	Chapel	Congress Centre	CETA	Train Station	Morgados	Museum	University
RMSE	18.09	15.16	16.46	16.7	15.21	21.07	12.59	17.74	15.17
NMB	−0.117	−0.32	−0.056	−0.166	−0.039	−0.034	−0.203	0.05	−0.36
r (Pearson)	0.6	0.61	0.57	0.6	0.68	0.56	0.73	0.66	0.62
	CO								
	Library	Firefighters	Chapel	Congress Centre	CETA	Train Station	Morgados	Museum	University
RMSE	119.2	292.72	397.16	119.03	100.44	117.91	105.62	130.88	139.34
NMB	−0.01	−0.86	0.081	−0.105	−0.031	−0.041	0.149	0.046	0.026
r (Pearson)	0.88	0.27	0.31	0.87	0.91	0.85	0.9	0.8	0.82

## Data Availability

The data that support the findings of this study are available from the corresponding author, Daniel Graça, on reasonable request.

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
