# Peer review of "Sensors Network as an Added Value for the Characterization of Spatial and Temporal Air Quality Patterns at the Urban Scale"

_sensors, 2023, doi:10.3390/s23041859_

Round 1

Reviewer 1 Report

Rows 194 - 196
It is reported that the sensors were compared with measurements
from reference instruments in co-location, to validate their performance. What were the comparison results - RMSE etc.?
The indices given in section 2.4 do not correspond to co-located measurements.

Rows 197 - 199
The wind sensor is located very low above the roof level, the measurement may be affected by leeward effects; should be mentioned in the text.

Row 442
What is figure 34?

Row 452
Figure 6 is missing in the text.

Row 464 - 465
Figure 7 refers to CO, not NO2

Rows 487 - 488
The patterns are practically the same in all season stations...
It should probably be "sensors stations"

Reviewer 2 Report

In this study the results obtained by a network composed of 9 low-cost air quality sensors (PM10, PM2.5, NOâ‚‚, O₃ and CO) distributed within the city of Aveiro, Portugal are presented. The measurement campaign was conducted from June 2020 to May 2021, and the temporal and spatial distributions of selected pollutants are examined using standard descriptive statistics. The data for PM10, NOâ‚‚ and CO from the sensor stations were compared to the measurement from the one reference stations.  It is concluded that the data provided could be useful for map the air quality in the urban scale.

Generally, the paper is interesting because the presented results are related to the important air pollution problem. However, there are many similar studies providing information on regional air quality based on low-cost sensors and using standard methodology and analysis

While the topic looks worthy of investigating, I do not find anything new in the paper that contributes to the current knowledge. In this paper commercially available sensors were used and standard descriptive analysis of the measurements was conducted. Standard hourly, daily, and monthly analysis for each pollutant is presented as a report.

Having in mind that the results should be of wider scientific interest the authors should clearly explain the novelty and describe the difference with respect to the many other similar studies that have already been published and conducted worldwide.

The results could be useful for local/regional population and policy makers. Please find below several suggestions that can be used for improving the manuscript:

Introduction section:

This section is too long and can be reduced without losing any information, for example some sentence should be shorten and rephrased e.g., lines 118-128.

 Material and Methods:

If the two referent measurement sites (Ílhavo and Estarreja ) were not appropriate for comparison in this study they can be omitted in figures and manuscript.

Table 3. Having in mind the accuracy of the sensors and obtained correlation and NMB with respect to referent station for PM10 and NO2, the reliability of the measurements is not obvious. This should be discussed in depth. Also, AQI index is calculated based on all available data PM10, PM2.5, NO₂, O₃ but data for PM2.5 and O₃ were not previously compared to the referent station and thus methodology applied is not consistent.

 The discussion in some point is general and subjective, more like an assumption without reference, for example

Line 415e 415: “The increase in July, is probably due to far-away wildfires” . It could be useful if some additional analysis is provided to support this statement – back trajectory analysis, satellite confirmation, wind directions…

Since there are a lot of data obtained using these network sensors with high temporal resolution, more sophisticated analysis relevant for wider scientific community could be done.

Some minor typing errors should be fixed: line 472, please check eq. (2),

Lines 452-453: There is no Figure 6 presented

Reviewer 3 Report

General comment:

I really enjoyed reading such a nice piece of good work! Undoubtedly the paper is within the scope of the journal and after a few corrections can be published. Figures are self-explainable, of good quality, and wisely chosen. As the paper is focused on LCSs use, which with all their disadvantages, are easily available for societies, and can give important information for public health, I would like to see in the paper examples of other studies around the EU and preferably world showing the success stories of this devices. I wish the authors all the best in their further study.

Specific comments:

Introduction

1. Line 105 - is it needed to place the link in the text?

2. Please provide information about air quality standards in Spain and describe the problem of air pollution in the country and/or region

1. Kindly add a short description in the introduction about one of the most significant advantages of LCS - high spatial coverage. Please refer to the paper from Krakow (the most polluted city in the EU with good air quality planning) where a network of 100 LCSs was used - look at the 2021 special issue Sensors for Air Quality Monitoring of the SENSORS MDPI journal for reference

Material and methods

1. Line 158 - 160 Please describe the localization selection in a more detailed way. What are the sources you mention? Please add a description of the topography and the sensor's high

2. Line 169 - what type of station is it? Automatic or manual?

3. Please add more information about data pre-processing -> preferably with some figures.

4. Please add a separate figure with time series showing the reference measurements and the measurements only from the closes lcs station. 

Results and conclusions

427 - 446 Please discuss your observation about higher concentration in particular months and hours in relation to the paper from Krakow. Besides obvious differences in climate, is possible to state a general observation useful for public health?

Round 2

Reviewer 2 Report

The revised version of the manuscript is improved, the authors reasonably addressed most of the previous suggestions and included some explanations. Additional information is given in Supp. Mat. as well.  Cluster analysis is added revealing some differences between measuring stations for better understanding of air pollution spatial variations. In conclusion, the limitation of the study and LCS network is stated.

I would suggest some minor technical corrections before publishing:

Please check capital letters in Figure 3 (steps titles…Daily, weekly, Monthly….)

Figure 10: Please increase font in figure,

Figure 10: dist(thedata) and hclust(*,”complete”) should be removed

Cluster distance method used should be added (Line 245)

Supplementary Material and Appendix should be mention in the main manuscript

Reviewer 3 Report

I’m satisfied with corrections made by Authors. They also provide comprehensive and complete replies to all my comments.

I can recommend the paper for publication.

I wish all the best!

Author Response

Thank you! Some ajustments in the english were made and minor errors were corrected.